Mitochondrial cytochrome c oxidase subunit I (COI) metabarcoding of Foraminifera communities using taxon-specific primers

Macher Jan-Niklas jan.macher@naturalis.nl 1
Bloska Dimitra Maria 1
Holzmann Maria 2
Girard Elsa B. 1 3
Pawlowski Jan 4
Renema Willem 1 3
1 Marine Biodiversity, Naturalis Biodiversity Center , Leiden , The Netherlands
2 Department of Genetics & Evolution, University of Geneva , Geneva , Switzerland
3 Department of Ecosystem & Landscape Dynamics, University of Amsterdam , Amsterdam , Netherlands
4 Laboratory of Paleoceanography, Institute of Oceanology Polish Academy of Sciences , Sopot , Poland
Waiho Khor
Electronic publication date: 2022 Sep 5
Publication date: 2022
Volume: 10
Electronic Location ID: e13952
Received 2022 Mar 11; Accepted 2022 Aug 5
Copyright: ©2022 Macher et al.
Copyright year: 2022
Copyright holder: Macher et al.
License: This is an open access article distributed under the terms of the Creative Commons Attribution License, which permits unrestricted use, distribution, reproduction and adaptation in any medium and for any purpose provided that it is properly attributed. For attribution, the original author(s), title, publication source (PeerJ) and either DOI or URL of the article must be cited.
License URL: https://creativecommons.org/licenses/by/4.0/

Keywords: Foraminifera, Metabarcoding, Beach, Community composition, Intertidal, Molecular biodiversity

Funding: BEN (Biodiversity-Ecology-Nature) T0206/37197/2021/kg Bauer-Hollmann Foundation This work was funded by a BEN (Biodiversity-Ecology-Nature) grant (Number T0206/37197/2021/kg) of the Bauer-Hollmann foundation to Jan-Niklas Macher. The funders had no role in study design, data collection and analysis, decision to publish, or preparation of the manuscript.

==============================
Foraminifera are a species-rich phylum of rhizarian protists that are highly abundant in most marine environments. Molecular methods such as metabarcoding have revealed a high, yet undescribed diversity of Foraminifera. However, so far only one molecular marker, the 18S ribosomal RNA, was available for metabarcoding studies on Foraminifera. Primers that allow amplification of foraminiferal mitochondrial cytochrome oxidase I (COI) and identification of Foraminifera species were recently published. Here we test the performance of these primers for the amplification of whole foraminiferal communities, and compare their performance to that of the highly degenerate LerayXT primers, which amplify the same COI region in a wide range of eukaryotes. We applied metabarcoding to 48 samples taken along three transects spanning a North Sea beach in the Netherlands from dunes to the low tide level, and analysed both sediment samples and meiofauna samples, which contained taxa between 42 µm and 1 mm in body size obtained by decantation from sand samples. We used single-cell metabarcoding (Girard et al., 2022) to generate a COI reference library containing 32 species of Foraminifera, and used this to taxonomically annotate our community metabarcoding data. Our analyses show that the highly degenerate LerayXT primers do not amplify Foraminifera, while the Foraminifera primers are highly Foraminifera- specific, with about 90% of reads assigned to Foraminifera and amplifying taxa from all major groups, i.e., monothalamids, Globothalamea, and Tubothalamea. We identified 176 Foraminifera ASVs and found a change in Foraminifera community composition along the beach transects from high tide to low tide level, and a dominance of single-chambered monothalamid Foraminifera. Our results highlight that COI metabarcoding can be a powerful tool for assessing Foraminiferal communities.

Introduction

Foraminifera are a species-rich phylum of rhizarian protists (Goldstein, 1999; Gupta, 2003; Burki et al., 2010) found mostly in marine environments. Studies on species diversity and ecology of Foraminifera are conducted to understand both past (Keller, 1983; Charnock & Jones, 1990; Scheibner, Speijer & Marzouk, 2005) and recent (Hallock et al., 2003; Murray, 2006; Nooijer et al., 2008; Pawlowski et al., 2014) ecosystems. Foraminifera can be identified on both morphological characters and molecular data, which allows deeper insight into their diversity (Pawlowski & Holzmann, 2014; Pawlowski, Lejzerowicz & Esling, 2014; Morard et al., 2015; Morard et al., 2016). Molecular work on foraminiferal species has provided insights into their phylogeny (Holzmann et al., 2001; Pawlowski & Holzmann, 2002; Darling & Wade, 2008; Pawlowski, Holzmann & Tyszka, 2013; Holzmann & Pawlowski, 2017) and led to the discovery of cryptic diversity in widely distributed morphospecies (Darling, Kucera & Wade, 2007; Morard et al., 2016; Prazeres et al., 2020; Macher et al., 2021a). Molecular metabarcoding, i.e., the amplification, sequencing and analysis of whole communities (Taberlet et al., 2012), is commonly applied to Foraminifera communities to detect species diversity in a wide range of ecosystems (Pawlowski, Lejzerowicz & Esling, 2014; Morard et al., 2019; Holzmann et al., 2021), for environmental impact assessments (Pawlowski et al., 2014; Laroche et al., 2016; Frontalini et al., 2018) and to study the ecology of Foraminifera (He et al., 2019; Chronopoulou et al., 2019; Greco, Morard & Kucera, 2020). Until recently, only one genetic marker, the nuclear 18S ribosomal RNA (the small subunit ribosomal ribonucleic acid; SSU), has been used for barcoding and metabarcoding of Foraminifera. The 18S rRNA allows identification of most foraminiferal species (Pawlowski & Holzmann, 2014; Morard et al., 2019), but some species show minimal or no variability (Schweizer et al., 2011; Borrelli et al., 2018) and others show hypervariability (Weber & Pawlowski, 2014; Morard et al., 2016; Prazeres et al., 2020), which can hamper species identification. To further advance molecular studies on Foraminifera, additional and easily obtainable molecular markers are desirable. This could potentially improve species identification, phylogenetic analyses and ecological studies. Recently, the first mitochondrial cytochrome c oxidase subunit I (COI) gene sequences of Foraminifera were published, together with primers which amplify a COI fragment (Macher et al., 2021b; Girard et al., 2022) overlapping the established Leray fragment, which is used in metabarcoding studies on a wide range of Eukaryota (Leray et al., 2013; Wangensteen et al., 2018). COI is a promising molecular marker for species identification in Foraminifera, but so far the newly developed COI primers were only applied to single foraminiferal specimens for the purpose of reference barcoding and assessment of marker variability (Macher et al., 2021b; Girard et al., 2022). Here we test whether the new Foraminifera COI primers can be used for metabarcoding of Foraminifera communities from environmental samples.

We applied metabarcoding to 48 samples taken along three intertidal transects spanning a North Sea beach in the Netherlands from dunes to the low tide line. We test taxon specificity of the Foraminifera primers, i.e., whether they amplify mostly Foraminifera or also a wide range of other taxa as reported for several degenerate COI metabarcoding primers (Weigand & Macher, 2018; Wangensteen et al., 2018). We compare the performance of the Foraminifera COI primers to that of the highly degenerate LerayXT primers, which are known to amplify COI of a wide range of eukaryotes and even prokaryotes (Collins et al., 2019). Second, we test whether the inferred community composition of Foraminifera is influenced by the sampling method (sediment core vs. meiofauna sample), and third, whether the metabarcoding data can be used to infer changes in foraminiferal communities along the beach transects from high tide to low tide line.

Materials & Methods

Field work

We took samples along three parallel transects (3 m apart) from the dunes to the low-tide line at Katwijk in the Netherlands (coordinates: 52.188479 N, 4.376894 E) on 13th August 2021 at low tide. We sampled eight sites along each of the transects: the first sample was taken at the foot of the dunes, above the high-tide line, the second sample halfway between the dunes and the high-tide line, and the remaining six samples equidistantly spaced from high-tide line to low-tide line (see Fig. 1). Since we took samples from three parallel transects, we sampled a total of 24 sites. Samples were taken and the beach state was recorded according to protocols for sampling the intertidal zone of sandy beaches, including measurement of the beach slope and assessing the breaker height (McLachlan, Defeo & Short, 2018).

We collected two types of samples per sampling site: one sediment core of 5 cm diameter and a length of 10 cm (volume ≈200 ml), and a second sediment core of 1 cm diameter and a length of 10 cm (volume ≈8 ml), with standard, sterile plastic syringes cut open at the front end. Immediately after sampling, we transferred the small (≈8 ml) sediment core to a 50 ml falcon tube and preserved the sample with 30 ml of 96% EtOH. We extracted meiofauna from the larger sediment core (≈200 ml) by adding 500 ml of MgCl2 solution to the sediment in a sterile bottle, which anaesthetises meiofauna and allows subsequent separation from the sediment by decantation (Somerfield, Warwick & Moens, 2005). After 5 min, samples in MgCL2 solution were carefully swirled and the supernatant containing the meiofauna decanted through a 1 mm and 41 um sieve cascade, as commonly done in beach meiofauna studies (Haenel et al., 2017; Martínez et al., 2020; Castro et al., 2021; Gielings et al., 2021). The meiofauna fraction retained on the 41um sieve was rinsed into sterile 15 ml Falcon tubes and preserved with 10 ml 96% EtOH. All samples were transported back to the Naturalis Biodiversity Centre laboratory and stored at −20 °C until further processing.

Figure 1 Profile of Katwijk beach, showing the eight sampling points along the transect from dunes to the low tide line.

AHW1-AHW2: Sampling sites above the high water line; S1–S6: Sampling sites in the intertidal area.

DNA extraction

Ethanol from Falcon tubes containing the meiofauna samples was evaporated at 50 °C overnight in a sterile warming cabinet, and we transferred the dried samples to 2 ml Eppendorf tubes. We extracted DNA with the Macherey Nagel NucleoSpin Soil kit (Macherey Nagel, Düren, Germany) according to the standard protocol including bead beating, but starting with an additional overnight Proteinase K digestion step (50 ul 250 µg/ml protK (Thermo Fisher Scientific, Waltham, USA) added to the lysis buffer provided with the kit) to improve cell lysis, as done in previous studies on sediment meiofauna (Weigand & Macher, 2018; Martínez et al., 2020). The sediment samples were transferred to sterile petri dishes and dried overnight at 50 °C in a sterile warming cabinet. After drying, we homogenised each sample with a sterile spatula, and a subsample of 0.5 g was transferred to a 2 ml Eppendorf tube. The extraction followed the protocol as described above for meiofauna samples.

Amplification of COI for community metabarcoding

We amplified both sediment and meiofauna samples with a two-step PCR protocol, with the commonly used LerayXT primers targeting a wide range of Eukaryota (Wangensteen et al., 2018; Collins et al., 2019), and the recently published Foraminifera COI primers (Foraminifera_COI_fwd1: 5′-GWGGWGTTAATGCTGGTYGAAC -3′, Foraminifera_COI_rev: 5′- RWRCTTCWGGATGWCTAAGARATC-3′) targeting the same COI region (Macher et al., 2021b) with an amplicon length of 310 to 320 base pairs. For the first PCR, each reaction contained 11.7ul mQ water, 2ul Qiagen CL buffer (10x; Qiagen, Hilden, Germany), 0.4 ul MgCL2 (25 mM; Qiagen), 0.8ul Bovine Serum Albumine (BSA, 10 mg/ml), 0.4 ul dNTPs (2.5 mM), 0.2ul Qiagen Taq (5U/ul), 1ul of each nextera-tailed primer (10 pMol/ul), and 2.5 ul of DNA template. PCR amplification was performed with 3 min of initial denaturation at 96 °C, 30 cycles of denaturation for 15 s at 96 °C, annealing at 50 °C for 30 s, extension for 40 s at 72 °C, followed by a final extension at 72 °C for 5 min. Three negative controls (Milli-Q water, Merck, Kenilworth, USA) were processed together with the samples to check for potential contamination. After the first PCR, samples were cleaned with AMPure beads (Beckman Coulter, Brea, United States) at a ratio of 0.9:1 according to protocol. For the second PCR, samples were amplified with individually tagged primers following the same protocol as above, using the PCR product from the first PCR as template and with PCR cycle number reduced to 10. We measured DNA concentrations using the TapeStation (Agilent Technologies. Santa Clara, CA, USA) with the High Sensitivity Kit and equimolarly pooled samples. The final library was cleaned with AMPure beads as described above and sent for sequencing on part of an Illumina MiSeq run (2 × 300 bp read length) at Baseclear (Leiden, The Netherlands).

Sequencing of Foraminifera reference species

We performed single-cell metabarcoding on 36 specimens from 32 morphospecies of Foraminifera as a reference for taxonomic annotation, since only a limited number of foraminiferal COI references are available in the NCBI reference databases to date (Macher et al., 2021b). The DNA was obtained from the existing Foraminifera DNA collection of the Department of Genetics and Evolution, University of Geneva, curated by the co-authors Maria Holzmann and Jan Pawlowski. We amplified and sequenced the specimens using the Foram_COI_fwd1/ Foram_COI_rev primers with the metabarcoding protocol described above, with the following differences in PCR protocol: The DNA template for the first PCR was 10x diluted, and we used 40 cycles of amplification for the first PCR and 8 cycles for the tagging PCR. We measured DNA concentrations using the TapeStation (Agilent Technologies, Santa Clara, USA) and equimolarly pooled samples. Sequencing was conducted on part of a Illumina MiSeq run (2 × 300 bp) at Baseclear, Leiden. Bioinformatic processing was performed with the settings described below for community metabarcoding data. We performed a contamination check using the ‘decontam’ pipeline (v.1.14.0) (Davis et al., 2018) in R (v.4.1.2) using the package ‘phyloseq’ (v.1.38.0) (McMurdie & Holmes, 2013). To remove any potential low-key contaminations not picked up by the ‘decontam’ pipeline, we further retained only those ASVs with a read abundance >50% per specimen to obtain reliable references. We added 44 COI sequences of 16 previously sequenced species (Macher et al., 2021b) to the reference database to increase taxonomic coverage. To further increase taxonomic coverage, we amplified and Sanger sequenced 21 specimens from six morphospecies, with the same primers and the protocol described for COI barcoding of Foraminifera (Macher et al., 2021b). We quality trimmed and aligned these sequences in MEGA 11 (Tamura, Stecher & Kumar, 2021). One specimen (Glabratella sp. 17919) was excluded from further analyses due to low read quality and the sequence being shorter than expected.

We aligned all reference sequences using MAFFT (v. 1.4.0) (Katoh et al., 2002) as implemented in Geneious (v2020.2) and translated the sequences using mitochondrial translation table 4 to check for potential stop codons that might indicate sequencing errors or NUMTs (Macher et al., 2021b; Girard et al., 2022). This led to the exclusion of the sequences of Psammophaga sp. 19297, Psammophaga sp. 19263 and Psammophaga sp. 19299, which showed internal stop codons. As an additional quality check and to assess whether the newly generated reference sequences cluster in the expected phylogenetic position of a species based on previous studies, we calculated a cladogram using FastTree (Price, Dehal & Arkin, 2010) as implemented in Geneious Prime (v. 2021.2). Taxonomic assignment of the reference sequences followed (Holzmann et al., 2022; Holzmann et al., 2021; Pawlowski, Holzmann & Tyszka, 2013; Holzmann & Pawlowski, 2017; Siemensma et al., 2017; Renema, 2018). Furthermore, as in previous studies (Macher et al., 2021b; Girard et al., 2022), we used the ASAP species delimitation algorithm (Puillandre, Brouillet & Achaz, 2021) with default Kimura K80 settings on the newly created reference database to test which genetic distance is appropriate for defining species based on the amplified COI fragment.

Bioinformatic processing of community metabarcoding data

We processed metabarcoding raw reads using the Galaxy platform (Afgan et al., 2018) following a standard workflow (Beentjes et al., 2019; Macher et al., 2020; Girard et al., 2022). The LerayXT and the Foraminifera dataset were processed separately. FLASH (v1.2.11) (Magoč & Salzberg, 2011) was used to merge reads with a minimum overlap of 150, a maximum overlap of 300, and a maximum mismatch ratio of 0.2 allowed. Non-merged reads were discarded. Cutadapt (v.2.8) (Martin, 2011) was used to trim primers (settings: both primers need to be present, minimum number of matching bases 10, maximum error rate 0.2). PrinSeq (v.0.20.4) (Schmieder & Edwards, 2011) was used to filter and trim sequences to 300 base pairs to remove reads that contain gaps or indels, which can be present due to sequencing errors, NUMTs or amplification of non-eukaryotic taxa (Wangensteen et al., 2018; Macher et al., 2018; Collins et al., 2019). UNOISE implemented in USEARCH (v10.0.240) (Edgar, 2016) was used for clustering of sequences into Amplicon Sequence Variants (ASVs) with thresholds of alpha = 4 and a minimum number of eight reads for the denoising approach, which is similar to previously tested settings (Turon et al., 2019). In order to remove potential spurious sequences present due to low-key contamination or tag switching (Schnell, Bohmann & Gilbert, 2015), we removed samples with less than 10,000 reads from the dataset, and retained only ASVs with more than 0.01% read abundance per sample.

We annotated ASVs to taxonomic names using the BOLDigger tool (v.1.2.6) (Buchner & Leese, 2020), which accesses both public and private references deposited in the Barcode of Life Database (Ratnasingham & Hebert, 2007). For taxonomic identification of Foraminifera, both the LerayXT and Foraminifera dataset were additionally searched against the newly constructed reference database of Foraminifera COI sequences described above using blastn (Altschul et al., 1990), with the algorithm set to retrieve annotations for sequences with a minimum of 80% sequence identity and 90% query coverage to a reference. The annotations were added to the taxonomy table obtained from BOLDigger. Based on the ASAP species delimitation results we applied to the newly created reference database, and previous results (Girard et al., 2022), we used a threshold of 99% identity for species level matches. For higher taxonomic ranks we used identity thresholds that are similar or slightly stricter than those previously used used for assigning taxonomic ranks in unknown eukaryotic communities (Holovachov, 2016; Elbrecht et al., 2017; Weigand & Macher, 2018): genus 96%; family 94%; order 90%; class 85%; phylum 80%.

Taxon specificity of Foram primers and ASV richness

We assessed the taxon-specificity of the COI primers by calculating the fraction of reads annotated to Foraminifera in the two datasets, i.e., samples amplified with the LerayXT primers and samples amplified with the Foraminifera COI primers, and in the two treatments per dataset, i.e., the sediment samples and the meiofauna samples. We calculated ASV richness of Foraminifera using the R package ‘vegan’ (v.2.5.7) (Oksanen et al., 2020).

Comparison of Foraminifera communities in sediment and meiofauna samples

For the comparison of Foraminifera community composition in sediment and meiofauna samples, we reduced the datasets to those 11 sampling sites for which we retained both sediment and meiofauna data after bioinformatic processing. We analysed the community composition using the ‘adonis’ PERMANOVA as implemented in the R package ‘vegan’, based on presence/absence of ASVs. The data was converted to presence/absence to avoid analyzing PCR biased data (Elbrecht & Leese, 2015; Leray & Knowlton, 2017). The factor ‘sample treatment’ (sediment sample vs. meiofauna sample) was used as the predictor and the Sørensen distances as response variable. Sørensen distances were calculated using the vegdist function implemented in the ‘vegan’ package. Communities were subsequently clustered with an average-linkage algorithm (hclust function) as in previous studies (Burdon et al., 2016; Macher et al., 2018). We regarded significant results with R2 >0.09 (equivalent to r = 0.30) as moderate, and R2 > 0.25 (r = 0.50) as strong (Nakagawa & Cuthill, 2007; Cohen, 2013). We visualised the number of ASVs found in both sediment and meiofauna samples and those found exclusively in one of the treatments with Venn diagrams using the R package ‘VennDiagram’ (v.1.7.1) (https://cran.r-project.org/package=VennDiagram).

Community composition of Foraminifera along the intertidal transect

The dataset used for analysis of the Foraminifera community along the intertidal transect comprised 18 sediment samples, i.e., six samples along each of the three parallel transects spanning the intertidal zone from high tide to low tide line. We did not analyse the LerayXT dataset further because no Foraminifera were identified in this dataset. We also did not further analyse the meiofauna samples amplified with the Foraminifera primers, as amplification and sequencing success was low for these samples.

To assess whether we captured the whole richness of Foraminifera ASVs present in the study sites, we applied interpolation and extrapolation analyses as implemented in the R package iNEXT (v.2.0.20) (Hsieh, Ma & Chao, 2016). A TukeyHSD test as implemented in R was used to test for differences in ASV richness between samples at different tidal levels. Both iNEXT and Tukey HSD tests were performed based on previously published scripts (Marwayana, Gold & Barber, 2021). We analysed the community composition of Foraminifera ASVs along the intertidal transect using the ‘adonis’ PERMANOVA as described above, with ‘transects’ (one to three) and ‘tidal level’ as predictors, and Bray-Curtis distances as response variables. For the comparison of community composition in different tidal levels, we categorised the sampling sites into ‘upper intertidal’ (sampling locations S1, S2; six samples), ‘middle intertidal’ (S3, S4; six samples) and ‘lower intertidal’ (S5, S6; six samples). Non-Metric Multi-Dimensional Scaling (NMDS) plots were calculated and visualised with the R libraries ‘vegan’ and ‘phyloseq’ (McMurdie & Holmes, 2013).

Results

Katwijk beach was classified as a tide modified beach with a reflective state based on beach morphology and the calculated Relative Tide Range (RTR) of 4.47 on the day of sampling. See Fig. 1 for the beach profile and the location of sampling points along the three parallel transects.

Amplification and sequencing of taxonomic references

We applied single-cell metabarcoding to 36 specimens from 32 morphologically identified Foraminifera species. The quality check with the ‘decontam’ pipeline showed none of the ASVs to be a potential contamination. For 26 out of 32 sequenced species, we identified a single, species-specific ASV sequence. The exceptions were Glabratellina sp. (isolate number 17921) and Planoglabratella opercularis (isolate number 18053; both in the family Glabratellidae), which shared the same ASV sequence, and the same was found for Trifarina earlandi (isolate number 17160) and Uvigerina bifurcata (isolate number 17173; both in the family Uvigerinidae). This was confirmed by the additional Sanger sequenced specimens of Trifarina, Uvigerina and Planoglabratella, which show the same sequence as the ASV sequences obtained through single-cell metabarcoding. The two Neoassilina ammonoides specimens showed a distinct ASV sequence per specimen. See Supplementary Material 1 for ASV sequences and read numbers per sample.

The cladogram shows that the newly generated reference sequences cluster in the expected higher taxonomic groups (the non-monophyletic monothalamids, and the classes Globothalamea and Tubothalamea), and in the previously defined superfamilies and clades (Pawlowski, Holzmann & Tyszka, 2013; Holzmann & Pawlowski, 2017; Holzmann et al., 2021). However, Cassidulinoides, Globocassidulina and Bolivina (Serioidea) do not cluster with the other Serioidea, but as a sister clade to Glabratelloidea, the remaining Serioidea, Globigerinoidea and Rotaloidea. Further, Stainforthia and Epistominella, which are assigned to the globothalamid “Clade 3”, do not cluster with the other “Clade 3” specimens, but with the Rotalioidea. See Fig. S1 for the cladogram.

The ASAP analysis delineated 50 lineages (ASAP score: 4.5, P-value: 2.67e–01, W: 3.96e–04) in the reference dataset and showed a best species delineation threshold of 0.82% genetic distance. Since previous ASAP analyses (Girard et al., 2022) showed that delineation thresholds can vary between taxonomic groups within Foraminifera, but are around 1% genetic distance, we further used 99% identity as the threshold for assigning sequences from the metabarcoding dataset to a species name.

Amplification and sequencing of community samples

Amplification of community samples was successful for 46 out of 48 samples with the LerayXT primers, and for 43 out of 48 samples using the Foraminifera COI primers. However, several meiofauna samples amplified with the Foraminifera COI primers as well as samples from above the high-tide line showed weak amplification, and these samples were subsequently lost during sequencing and bioinformatic processing. See below for details. The negative controls did not show any product (as tested by TapeStation with High Sensitivity Kit) and were therefore not sequenced. Sequencing resulted in 5,849,271 raw reads for the LerayXT sample, and 2,431,554 raw reads for the Foraminifera samples. After bioinformatic processing, the final LerayXT dataset comprised 2,963,461 merged and quality filtered sequences, and the Foraminifera dataset comprised 1,097,955 sequences. We retained 43 samples for the LerayXT dataset (22 meiofauna samples, 21 sediment samples) containing 4025 ASVs, and 34 samples for the Foraminifera dataset (14 meiofauna samples and 20 sediment samples) containing 763 ASVs. See Supplementary Material 2 for read number per sample and nucleotide sequences.

Taxon- specificity of Foraminifera primers

Taxonomic annotation using Barcode of Life (BOLD) and the newly generated Foraminifera COI reference database showed that none of the reads in the LerayXT dataset were annotated to a Foraminifera reference sequence. In contrast, 176 ASVs out of the 763 ASVs were identified as Foraminifera in the samples amplified with the Foraminifera COI primers.

In the LerayXT-amplified meiofauna samples, most reads were annotated to Annelida (48%) and Arthropoda (28%), while 14% could not be assigned to a phylum. In contrast, 88.6% of reads in the Foraminifera meiofauna dataset were assigned to Foraminifera, and 9% could not be assigned to a taxonomic group. Other taxa (mostly: Annelida and Arthropoda) were present with less than 2% of reads. See Figs. 2A and 2B; and see Table S2 for a complete list of taxa.

Figure 2 Pie charts showing the proportion of reads annotated on phylum level.

(A) LerayXT meiofauna samples, (B) Foraminifera meiofauna samples, (C) lerayXT sediment samples, (D) foraminifera sediment samples. Phyla present with <2% of annotated reads are summarised as “other taxa’ to improve readability.

In the LerayXT- amplified sediment samples, the largest fraction of reads (28%) could not be assigned to a phylum, 22% were assigned to Bacillariophyta (diatoms), and 19% to Arthropoda. In the sediment samples amplified with the Foraminifera primers, 88.9% of reads were annotated to a Foraminifera reference, and 9% were not assigned to a phylum. Other taxa were present with less than 2% of all reads. See Figs. 2C and 2D.

The majority of Foraminifera ASVs could not be identified on species level, but only on Class or Order level (see Table 1). The 138 Foraminifera ASVs that could be identified at least on class level belonged to the monothalamids, globothalamids and tubothalamids. See Table 2.

Comparison of Foraminifera communities in sediment and meiofauna samples

We retained 22 samples (11 meiofauna, 11 sediment) for the comparison of Foraminifera communities in sediment and meiofauna samples. These samples contained 154 foraminiferal ASVs. 71 ASVs (49.6%) were found in both sample types, 49 ASVs (34.3%) were exclusively found in the sediment samples, and 23 ASVs (16.1%) were only found in meiofauna samples (Fig. 3). The ‘adonis’ test implemented in vegan showed that ASV community composition differed significantly between meiofauna and sediment samples (R2: 0.184, p = 0.001∗∗∗). ASVs assigned to Tubothalamea were found only in the meiofauna samples. Globothalamea ASVs were most common in meiofauna samples, while monothalamid ASVs were most common in sediment samples. See Table 3.

Table 1 Number and percentage of foraminifera ASVs identified on phylum, class, family, genus and species level.

Taxonomic level	ASVs	
Phylum level	176 (100%)	
Class level	138 (78.41%)	
Order level	116 (66.48%)	
Family level	69 (39.2%)	
Genus level	53 (30.11%)	
Species level	9 (5.11%)	

Table 2 Number and percentage of Foraminifera ASVs assigned to monothalamids, globothalemea and tubothalamea.

Taxonomic group	ASVs	
Monothalamids	68 (49.28%)	
Globothalamea	65 (47.1%)	
Tubothalamea	5 (3.62%)	

Figure 3 Venn diagrams showing Foraminifera ASVs exclusively found in either the sediment or meiofauna samples, and ASVs found with both sampling techniques.

Foraminifera community composition along the intertidal transect

The final ‘Foraminifera sediment’ dataset for the analysis of community composition along the intertidal transect comprised 18 samples with 149 Foraminifera ASVs. The dataset comprised samples from six tidal levels (S1- high tide line, to S6- low tide line) with three biological replicates each. ASV richness did not differ significantly between tidal levels (TukeyHSD test results: Supplementary Material 3). Inter- and extrapolation showed that ASV richness approached saturation, with a sample coverage of 92.5%. See Fig. S2.

Analysis of community composition with ‘adonis’ PERMANOVA showed that Foraminifera ASV communities did not differ significantly between transects (R2: 0.087, p = 0.954), but moderately between tidal levels (R2: 0.187, p = 0.005∗). NMDS plots (stress: 0.186) show that community composition in ‘upper intertidal’ samples differed the most from community composition in samples from the ‘middle’ and ‘lower intertidal’ area (Fig. 4). Communities in samples from the upper intertidal do not cluster together, showing heterogeneity of community composition, while samples from middle and lower intertidal cluster together. The majority of ASVs in the upper intertidal zone had a taxonomic match to monothalamids (59.81%). In the middle intertidal zone, 68.29% of ASVs were annotated to monothalamids, and in the lower intertidal zone, 75.29% of ASVs were annotated to monothalamids.

Table 3 Number and percentage of Foraminifera ASVs identified on class level in sediment and meiofauna samples.

	Meiofauna samples	Sediment samples	
Taxonomic Group	No. of ASVs	No. of ASVs	
Monothalamids	25 (31.25%)	55 (56.07%)	
Globothalamea	50 (62.5%)	42 (43.3%)	
Tubothalamea	5 (6.25%)	/	

Figure 4 NMDS plot showing similarity of Foraminifera communities.

NMDS plot showing similarity of Foraminifera communities (based on ASVs) between samples from upper, middle and lower intertidal sampling sites. Shapes and colours show the sampled tidal levels. Stress = 0.19.

Discussion

We report the first application of mitochondrial COI metabarcoding to infer Foraminifera communities from environmental and bulk samples, and show that the recently published Foraminifera COI primers amplify a wide range of foraminiferal taxa. The Foraminifera COI primers are highly taxon-specific, and we show that COI metabarcoding of Foraminifera can be used to infer changes in community composition along an intertidal transect. The availability of an easily obtainable molecular marker additionally to the commonly used 18S rRNA for metabarcoding of Foraminifera will be valuable for future studies, as it will open up new possibilities for studies on diversity and ecology of Foraminifera, and allows including multiple markers into molecular studies. This is becoming more common in metabarcoding studies (Dupuis, Roe & Sperling, 2012; Gao et al., 2014; Fais et al., 2020; Eberle et al., 2020; Gielings et al., 2021) and could improve studies on phylogeny, species diversity and ecology of Foraminifera.

Amplification and sequencing of Foraminifera COI reference sequences

The calculated cladogram showed that all 49 newly generated reference sequences cluster in the expected taxonomic groups (monothalamids, and the classes Globothalamea and Tubothalamea). However, we point out that we cannot and do not want to make definite statements on the phylogeny of Foraminifera based on the analysed, short COI marker, since we calculated the cladogram as a quality check and are aware that short markers can result in weakly supported phylogenies. While the majority of species show a species-specific COI sequence, we note some exceptions where different species or genera share the same COI sequence, as seen in Psammophaga spp. (isolates 19260, 19296), Glabratellina sp. and Planoglabratella opercularis, and Trifarina earlandi and Uvigerina bifurcata. This indicates that the short COI marker used here does not always differentiate between closely related species or even genera. On the other hand, closely related species like the two Rosalina spp., which show distinct 18S rRNA based on previously published data, also show distinct COI sequences, and some species such as Neoassilina ammonoides show a high variability in their COI sequences. This has been documented before in Foraminifera (Girard et al., 2022), and similar discordance of morphological species identification and COI barcodes due to low variability of the barcoding region have been reported from a range of other taxonomic groups, e.g., Amoebozoa (Tekle, 2014), arthropods (Havemann et al., 2018) and Porifera (Yakhnenko & Itskovich, 2019). Further research based on a higher number of single-cell (meta)barcoded specimens per well-identified morphospecies is therefore needed to investigate how reliable and consistently the COI metabarcoding marker can distinguish different closely related species or genera, and which species the marker might not resolve.

Amplification and sequencing of Foraminifera communities

We successfully amplified and sequenced DNA from intertidal meiofauna and sediment samples using both LerayXT and Foraminifera COI samples. However, amplification and sequencing success differed between primers, and between meiofauna and sediment samples. None of the reads in the LerayXT datasets was assigned to Foraminifera. While these primers were originally designed and are commonly used for metabarcoding metazoan taxa, they are known to amplify most main groups of Eukaryota and even prokaryotes (Wangensteen et al., 2018; Garcés-Pastor et al., 2019). The lack of Foraminifera sequences in this dataset therefore hints at a divergence of the primer binding sites in Foraminifera, hampering amplification. Similar results have been reported for primers targeting 18S rRNA, and subsequently Foraminifera-specific 18S primer combinations were developed (de Vargas et al., 1997; Pawlowski, 2000; Morard et al., 2011). When applying the Foraminifera COI primers, we retained 11 out of 24 meiofauna samples, and 20 out of 24 sediment samples. We speculate that the relatively low amplification and sequencing success observed for the meiofauna samples amplified with Foraminifera primers might be due to the lower abundance of Foraminifera in these samples. This seems possible as previous studies on intertidal Foraminifera of the North Sea coast reported relatively low abundances of sometimes less than 10 ind/cm3, (Horton, Edwards & Lloyd, 1999; Müller-Navarra, Milker & Schmiedl, 2016), while metazoan meiofaunal taxa like polychaetes can reach abundances of several thousand specimens per 10 cm2 in North Sea beaches (Kotwicki et al., 2005). Thereby, the ratio of target DNA (Foraminifera) to non-target DNA (metazoan taxa) might have been unfavorable in the meiofauna samples. However, other studies on intertidal Foraminifera based on morphological analyses reported a high abundance, but also high variability between sites (Reiter, 1959; Kameswara & Srinath, 2002; Lübbers & Schönfeld, 2018), and higher abundances of Foraminifera are mostly reported from muddy sediments (Morvan et al., 2006; Papaspyrou et al., 2013). It is also possible that the MgCl2 decantation method does not efficiently separate Foraminifera from sediment due to the calcareous test of many species, or that further amplification protocol optimisations like DNA dilutions are needed. In future studies, a size selection targeting smaller organisms, as used in previous studies on Foraminifera (Peeters et al., 1999; Langezaal et al., 2003; Pawlowski et al., 2005), could be applied to potentially increase the ratio of Foraminifera to metazoan meiofauna, and metabarcoding results should be compared to morphological analysis of samples.

Taxon-specificity of Foraminifera primers

We show that the majority of reads in both meiofauna and sediment samples amplified with the Foraminifera COI primers could be assigned to a foraminiferal reference, with about 90% of reads assigned to Foraminifera. A further 9% of reads could not be assigned to a taxonomic group. It seems possible that these reads are either foraminiferal taxa without available COI references, sequencing artifacts, random amplification of genomic regions, or nuclear mitochondrial DNA (NUMTs) (Porter & Hajibabaei, 2021; Andújar et al., 2021; Graham, Gillespie & Krehenwinkel, 2021). With growing COI reference for Foraminifera and understanding of their mitochondrial genomes, future studies could implement NUMT removal pipelines like METAMATE (Andújar et al., 2021) and defining thresholds for assigning COI sequences in metabarcoding datasets to taxonomic levels could be established. In this study, we applied ASAP delineation to the newly generated Foraminifera COI reference database and found that a threshold of about 1% genetic distance is appropriate for delineating species in Foraminifera COI datasets, which is in line with previous results (Macher et al., 2021b; Girard et al., 2022). However, more data on a wider range of species, genera and families is needed to define thresholds for higher taxonomic ranks, and we stress that the standard values we use in the present study, which stem from previous studies on eukaryotic COI, might actually over- or underestimate the number of taxa in Foraminifera metabarcoding datasets. Future studies should address this question by studying the diversity of foraminiferal COI, which could eventually lead to establishing a delineation system similar to that for 18S rRNA introduced by (Morard et al., 2016).

We also point out that even though our results show that the Foraminifera COI primers amplify a wide range of Foraminifera from all main taxonomic groups, it remains unknown whether primers preferentially amplify certain Foraminifera taxa, thereby leading to primer bias as reported for other primers and taxonomic groups (Tedersoo et al., 2015; Pawluczyk et al., 2015; Elbrecht & Leese, 2015). Future studies should test this using mock communities of known taxonomic composition (Smith et al., 2017; Lamb et al., 2019), and by comparing COI marker to 18S rRNA marker data. Further, future studies should also test amplification of Foraminifera COI from other environments, e.g., from plankton samples (Morard et al., 2015), soil (Lejzerowicz et al., 2010), mudflats (Papaspyrou et al., 2013) or tropical reefs (Förderer, Rödder & Langer, 2018) to test the amplification success for a wide range of sample types.

Comparison of Foraminifera communities in sediment and meiofauna samples

We found that the inferred community composition of Foraminifera differed significantly between meiofauna and sediment samples. Tubothalamea, albeit rare, were exclusively found in meiofauna samples. Monothalamids were more common in sediment samples. Previous studies based on 18S rRNA metabarcoding showed that sediment samples routinely contain a high number of monothalamids that can often not be reliably assigned to a lower taxonomic level (Rodrigues et al., 2021; Nguyen et al., 2021), which is in line with our results. We speculate that more small, amoeboid monothalamid taxa were lost during sieving and decantation of the meiofauna fraction, and that they were therefore present in higher abundance in the sediment samples. Corresponding to this, previous studies commonly reported differences in inferred community composition when targeting the same taxonomic groups with different sampling methods (Brannock & Halanych, 2015; Macher et al., 2018; Holman et al., 2019; Castro et al., 2021). Choosing the sampling and processing method should therefore be based on the targeted taxonomic group in future studies.

Foraminifera community composition along the intertidal transect

We show that the inferred community composition of Foraminifera in sediment samples changes along a transect from high tide line to low tide line. The Foraminifera communities in upper, middle and lower intertidal were dominated by monothalamids, but the fraction of monothalamids increased from the upper to the lower intertidal area. Since the available COI database of Foraminifera is still limited and the majority of ASVs could not be taxonomically assigned on family, genus or species level, more specific analyses on community composition are not possible until the number of COI reference barcodes increases. The increase in monothalamids towards the low-tide line might indicate that the environmental conditions (among others: sediment covered by seawater for longer time, more stable conditions due to lower wave energy, increased sediment grain size) might favor monothalamids over globothalameans and tubothalameans. Our findings are in line with previous studies based on morphology that show a change in intertidal Foraminifera communities corresponding to tidal levels (Horton, 1999; Horton & Culver, 2008; Rush et al., 2021), but we point out the need for more studies on Foraminifera COI metabarcoding to assess how meaningful ecological patterns can be extracted. Future studies should also measure more parameters like sediment grain size, salinity, oxygen content and nutrient content of the sediment. Metabarcoding studies based on 18S rRNA have shown the versatility of the approach for studying foram community changes in different environments (Pawlowski et al., 2014; Frontalini et al., 2018; Frontalini et al., 2020; Cordier et al., 2019).

We cannot exclude that some of the found DNA is environmental DNA not usually live in the sandy beach ecosystem, as has been shown for many testate (Taberlet et al., 2012) or stems from Foraminifera that were washed ashore, but do Foraminifera (Murray, 2006). However, we point out that monothalamids, which were the most common taxa in our samples, have been regularly found in intertidal sandy beach ecosystems (Larkin & Gooday, 2004; Golemansky, 2007; Alvarado & Goti, 2019). Future studies should also compare the established 18S rRNA metabarcoding approach for Foraminifera to COI metabarcoding to assess how comparable the results are for inferring community composition. Furthermore, reference barcoding of species using both 18S and COI should be continued (Pawlowski & Holzmann, 2014; Macher et al., 2021b; Girard et al., 2022). To allow studying North Sea beach ecosystems in more detail, targeted barcoding of taxa known to live in intertidal habitats of Northern European coats should be performed, e.g., more species of Haynesina, Ammonia and Elphidium (Alve, 2001; Brouwer et al., 2015). Since integrative approaches revealed a high diversity of yet unidentified Foraminifera and especially monothalamids (Voltski & Pawlowski, 2015), this taxonomic group needs special attention to fill gaps in reference libraries.

Conclusions

Our study shows that COI metabarcoding of foraminiferal communities with highly taxon-specific primers is feasible, thereby making mitochondrial metabarcoding available for further studies on Foraminifera. Continuing the build-up of Foraminifera COI reference databases based on morphologically identified species will be crucial to allow for species identification and refined ecological studies. Our results highlight that COI metabarcoding can be a powerful tool for assessing foraminiferal communities.

Supplemental Information

Supplemental Information 1 Foraminifera single-cell reference metabarcoding, quality filtered ASV table after 50% abundance subsetting

Click here for additional data file.

Supplemental Information 2 Raw reads and quality filtered reads for both LerayXT and Foraminifera datasets

Click here for additional data file.

Supplemental Information 3 TukeyHSD test results showing differences in ASV richness between intertidal sampling sites

Click here for additional data file.

Supplemental Information 4 Cladogram showing phylogenetic placement of sequenced Foraminifera specimens

Click here for additional data file.

Supplemental Information 5 Accumulation curves showing ASV richness in sediment samples

Fig. S2 Accumulation curves showing ASV richness in sediment samples amplified with Foraminifera COI primers. Solid lines show interpolated data, dashed lines show extrapolated data.

Click here for additional data file.

Supplemental Information 6 Foraminifere reference sequences submitted to GenBank, accession numbers OM719617 –OM719669, for review

Click here for additional data file.

We thank Elza Duijm, Marina Ventayol Garcia, Marcel Eurlings, Roland Butôt and Julide Cankat for help with laboratory work.

Additional Information and Declarations

Competing Interests

Author Contributions

Data Availability

The authors declare there are no competing interests.

Jan-Niklas Macher conceived and designed the experiments, performed the experiments, analyzed the data, prepared figures and/or tables, authored or reviewed drafts of the article, and approved the final draft.

Dimitra Maria Bloska conceived and designed the experiments, performed the experiments, analyzed the data, prepared figures and/or tables, authored or reviewed drafts of the article, and approved the final draft.

Maria Holzmann conceived and designed the experiments, performed the experiments, analyzed the data, authored or reviewed drafts of the article, and approved the final draft.

Elsa B. Girard conceived and designed the experiments, performed the experiments, analyzed the data, authored or reviewed drafts of the article, and approved the final draft.

Jan Pawlowski conceived and designed the experiments, analyzed the data, authored or reviewed drafts of the article, and approved the final draft.

Willem Renema conceived and designed the experiments, analyzed the data, authored or reviewed drafts of the article, and approved the final draft.

The following information was supplied regarding data availability:

All COI reference sequences are available in NCBI GenBank: OM719617 –OM719669.

The metabarcoding raw data are available in NCBI Sequence Read Archive: PRJNA799248.

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
