# Peer review of "Mitochondrial cytochrome c oxidase subunit I (COI) metabarcoding of Foraminifera communities using taxon-specific primers"

_PeerJ, doi:10.7717/peerj.13952_

## Round 0.1 · original submission · Major Revisions

The comments from the reviewers have encompassed all my concerns. Kindly respond to them accordingly and I look forward to reading the revised version of this manuscript.

Reviewer 1 ·

Basic reporting

.

Experimental design

.

Validity of the findings

.

Additional comments

The manuscript of Macher and colleagues compares newly developed foramifera-specific PCR primers targeting a mitochondrial gene (COI) to the primers amplifying eukaryotes that are commonly used for metabarcoding of communities from environmental samples (the Leray fragment). Because COI databases are extremely scarce in foraminifera barcodes, the authors have sequenced individual morphotypes (both sanger and metabarcoding) to i) develop a custom forams reference database for the COI marker and ii) asses the level of intra / inter species variability.
Using intertidal transects in a beach of the North Sea as test case, the results indicate that the developed forams-specific COI primers can deliver meaningful ecological data to address ecological questions on benthic foraminifera communities, whereas the Leray primers cannot seem to capture any foraminifera read.

The methods and results are convincing and indeed pave the way for further work with foraminifera COI marker, I have only a few minor comments.

The fact that distant genera can have the exact same COI sequence in the targeted marker and that some morphotype have multiple haplotypes seems limiting to advocate the use of the COI marker for ecological work. Even though the results section stipulates these (L.273-281), the discussion only propose to increase the number of species/morphotypes analyzed to better document this (non)variation and identify which taxa could be problematic with the COI marker. Although only a limited number of the tested species/morphotypes had such an issue, I think this would deserve a bit more attention in the paper. Why distant genera would share the same COI sequence? Has “intragenomic” variation of COI been documented in other taxonomic groups?

L.169: ASVs present in single morphotypes and that are representing less than 50% of the reads are discarded. Could you also document all the ASVs in the supplementary? Are they very distant sequences? Are they artefacts?

L.224: Dixon 2003 is the first citation for the vegan R package, but the recent should be preferred; since it will include the package version too, by typing in R the following:
citation(“vegan”)

L.231-233: If only presence absence data has been analyzed, this is not Bray-Curtis dissimilarity, but the Sørensen dissimilarity metric. Please clarify. Could you also justify why you would analyze only presence absence?

Abstract; could you briefly introduce the “meiofauna sample”? What is meant by this may be unclear to some readers.

L.180: Could you please introduce the NUMTs acronym in the manuscript?

Reviewer 2 ·

Basic reporting

This paper is on a very interesting topic, the development of a new marker to study the diversity of foraminifera and its application. Unfortunately, the reference database does not contain sequences of the species found in the studied area. As the authors sequenced new specimens, it would have been nice to target more specifically the diversity found in European intertital areas (which is well known and sequenced). A second problem is the sampled area, a sandy beach, where hard shelled foraminifera usually do not live. The samples are therefore certainly not representative of an abundant live community and most probably contained propagules and/or DNA of expatriated forams instead. To prove me wrong, you need to look at stained (Rose Bengal or CTG) foraminifera from the same area to show that live adults could be present there.
For the three aims of the paper, the taxon specificity of the primers was well demonstrated. The community composition change due to the sampling method is also displayed clearly but was already known from former studies (references are needed here). It was also shown previously with the SSU marker that metabarcoding data can be used to infer foram communities changes in different environments. To compare the new COI marker with the SSU marker would be more convincing, but you need to be sure that you work with a real foram community.

The literature on forams lacks references, particularly more general ones (details given in additional comments) and is biased toward planktonic and tropical foraminifera, although the topic of the paper is benthic forams from a temperate intertidal zone.

Raw data : I did not see any indication that the sequences were submitted to GenBank (for single cell data) or another repository for HTS data.

Experimental design

It would have been interesting to compare COI with SSU metabarcoding to compare performances of both markers on the same community as SSU is the only marker used with forams for the moment.
The sampled area seems to be a sandbeach. Could you precise the granulometry of the sediment sampled in the different transects? Have you checked with morphological approaches that forams actually live in this area ? Usually, shelled forams (Globothalamea and Tubothalamea) will avoid large particles of sediment and prefer to live in mudflats (see Murray 2006 for example). There is a lack of data for monothalamids.
It is sometimes hard to follow between the different extraction methods and primer sets used in the study. As the primer set LerayXT did not give any foram sequence, it may be easier to just mention it briefly somewhere (results ?) but not describe it in length in the M&M and result sections.

Validity of the findings

no comment

Additional comments

l. 23 : most instead of many
l. 32 : analysed instead as you use British English in the rest of the manuscript
l. 33 : single-cell DNA barcoding instead (metabarcoding is for communities
l. 36 : taxon-specific -> could you specify at which taxonomic level ?
l. 45 : other refs ?
l. 47-48 : Better to choose references more connected to the studied location, e.g. European studies, or more general ones, e.g. Murray’s books on ecology of forams.
l. 51 : foraminiferal species instead
l.52-53 : More general papers needed here, e.g. Pawlowski et al. 2013, Darling, K. F., & Wade, C. M. (2008). The genetic diversity of planktic foraminifera and the global distribution of ribosomal RNA genotypes. Marine Micropaleontology, 67(3-4), 216-238, instead of Darling et al. 1996 and Holzmann et al. 2001
l. 61 : As the SSU rDNA of forams is longer than in other eukaryotes, its value is probably not 18S, it may be better to use SSU instead.
l. 62-64 : There is one known example where the SSU marker cannot distinguish two morphospecies of forams, in Schweizer, M., Fontaine, D., & Pawlowski, J. (2011). Phylogenetic position of two Patagonian Cibicididae (Rotaliida, foraminifera): Cibicidoides dispars (d’Orbigny, 1839) and Cibicidoides variabilis (d’Orbigny, 1826). Revue de micropaléontologie, 54(3), 175-182. A good reference for intraindividual variability is Weber, A. A. T., & Pawlowski, J. (2014). Wide occurrence of SSU rDNA intragenomic polymorphism in foraminifera and its implications for molecular species identification. Protist, 165(5), 645-661.
l. 65-66 : Specify better what are the positive points of having different markers.
l. 77 : Were dunes above the intertidal zone ?
l. 81 : sampling method instead
l. 90-91 : When did you sample, at low tide or different times ?
l. 93-95 : Did you measure the granulometry of the different samples ?
l. 106 : EtOH is not ideal to preserve foraminiferal DNA (ask your co-authors Maria and Jan about that).
l. 120 : What is the dry weight of meiofaunal samples, is it comparable to the 0.5g of the sediment samples ?
l. 131 Twice « both » in the same line.
l. 174 : What do you mean by curate ? Could you be more specific ?
l. 182-185 : Did you also use SSU sequences of the same individuals to check the phylogeny ? As the SSU marker is widely used, it would be a more effective method to check the quality of the new COI marker.
l. 194-196 : Reference for Cutadapt ?
l. 198 : non-eukaryotic instead
l. 213 : It would be good to see this table in supplementary data.
l. 245-246 : Is it then useful to speak about this dataset before ?
l. 269 : Are they the species from the ref sequences fasta file ? A table with the species list would be useful. Typical intertidal genera such as Ammonia are missing from this database. Did you submit these sequences to GenBank ?
l. 292-304 : No need to present the LerayXT results as no foram read was obtained. Were these sequences deposited in a public repository ?
l. 293-294 : As no foram was sequenced with LerayXT primers, it is best to not speak about them anymore (this paper is focusing on forams).
l. 297-298 : Did you check that on the gel or by sequencing ?
l. 309-310 : Out of how many ASVs ?
l. 314 : What were the other taxa ?
l. 321-322 : This is certainly due to the fact that typical forams from the studied area are not represented in the reference database. The database is not very useful if you cannot identify below class level…
l. 335-337 : Interesting result, any explanation ?
l. 366-367 : Could you list what are the advantages of multiple markers ?
l. 374 : cannot instead
l. 374-375 : You could also say that the phylogenetic signal is probably weak with such a short marker.
l. 380 : closely related species instead
l. 380 : You do not need to repeat that this marker is short, actually it is longer that the SSU fragment used for many eDNA studies(<200nt).
l. 384 : DNA barcoded instead (metabarcoding is for communities, not individuals)
l. 386 : closely related species instead
l. 399-401 : The low abundance of forams is probably due to the fact that the sampled area was not ideal for the hard shelled forams. You may have sequenced propagules or free DNA instead of forams living actually there.
l. 403 : Alve & Murray 2001, this study is in South England in the Channel, not the North Sea. In mudflats, the density of forams varies during the year, but can be very high. Better to find other references in the North Sea.
l. 409-410 : Good argument
l. 415 : This is also important as I am not convinced that many forams lived in your sampled sandy area…
l. 432 : rDNA data as you study DNA and not RNA.
l. 442 : In your case, the whole set of sequences could not be assigned to lower taxonomic levels…
l. 449-450 : This is an important statement, which should always be kept in mind when designing sampling
l. 459-462 : You cannot make this conclusion as it is not sure that Globothalamea and Tubothalamea lived in this environnement.
l. 463-465 : Not possible either to conclude with your dataset.
l. 467-469 : This would have been more interesting to already compare both markers on these samples within the present study if you wanted to discuss your third aim.
l. 469-473 :
I totally agree with your conclusion.

Reviewer 3 ·

Basic reporting

Jan-Niklas Macher and coworkers present here the first metabarcoding study using the new developed specific primers for foraminifera targeting the COI gene. This paper is a follow-up from previous publications where the primers were described and validated on single cells. They have tested their approach on three transects from dune to low tide. They have compared two different methods of sampling (meiofauna and sediment) and two primers sets (general and specific). To increase the reference database, they have also sequenced single cells. They have highlighted the efficiency of the specific primers and conclude that the use of specific COI primers will be a powerful tool for assessing Foraminiferal communities. The article is well written, and the experimental design is correct to answer the scientific questions. Nevertheless, I have noticed several points that deserved specific attention from the authors before a potential acceptation for publication in PeerJ.

As this paper aim to present a new method to analyze foraminiferal communities, a specific attention must be taken to the details as future studies will potentially based all their methods on this paper.

In Line 214 the authors define thresholds to set up the different taxonomical ranks (species 98%; genus 96%; family 94%; order 90%; class 85%; phylum 80%) and cite three references to justify this choice. The authors assumed that for Foraminifera these thresholds will also be suitable. I disagree with this choice as it was demonstrated many times that the COI barcoding gap can vary between and across the tree of life. For example, it was shown that the barcoding gap in the COI gene for species in amoeba are 4 percent (Kosakyan et al., 2012). Therefore, without a proper analysis to see if the thresholds proposed by the references are also valid and adapted for Foraminifera, the proposed thresholds must be taken with high cautious as it will strongly influence the output of the dataset. I suggest to tone down this part or provide support for the validity of theses thresholds to avoid future publication based on these arbitrary values.

The advantage of eDNA metabarcoding studies are the ability to sequence environmental clades. In their study, the authors have assigned to species level 11.93% of the ASVs. Meaning that the large part of the diversity remains unassigned due to the current lack of single cell sequencing. So, I would like to know more of this aspect related to the COI gene. Do the authors expect to find a large proportion of cryptic diversity? Or do they assume that this unassigned diversity belongs to others species that are not yet barcoded? Did the authors apply the ASAP and ABGD algorithms used in their previous publications with the eDNA dataset to assess the species delimitation? This will also potentially provide useful information to define the species threshold.

In order to be able to redo the study, the authors must deposit their eDNA dataset in a public depository. I also recommend depositing the code used to do the bioinformatic and statistics online in a depository like Github. This will provide more strength to the study as everybody will be able to use more easily the protocol developed in this study.

Minor comments
- The captions of the figures and tables are missing (only Fig 1 has a caption)
- In the references section all the species and genera names must be in italic
- The manuscript contains a lot of “double space” between words. Please check the typos
- Specify each version of the softwares and the R package used in the manuscript
- Figure 2) Pie chart is a bad way to present community composition especially if you want to compare samples/treatments. I recommend using barplots.
- Fig 4) Value of stress is missing
- Line 95; typos “bbreaker”
- Line 135: Specify the length of the expected fragment

Kosakyan, A., Heger, T.J., Leander, B.S., Todorov, M., Mitchell, E.A.D., and Lara, E. (2012) COI barcoding of Nebelid testate amoebae (Amoebozoa: Arcellinida): extensive cryptic diversity and redefinition of the Hyalospheniidae Schultze. Protist 163: 415–434.

Experimental design

no comment: see basic report

Validity of the findings

no comment: see basic report

Additional comments

no comment: see basic report

---

## Round 0.2 · Minor Revisions

I agree with the reviewers that the current manuscript require just some minor revisions before acceptance. I look forward to your revision.

Reviewer 2 ·

Basic reporting

The authors answered most of my concerns. Below are some last comments:
l. 462-467: It is not so surprising that COI eukaryotic primers do not amplify foram DNA, as the same situation is observed with SSU universal eukaryotic primers which hardly amplify foram DNA even when it is in high quantities. You may add the SSU example in your discussion there.
l. 471-488: This low abundance of forams is certainly due to the sampling site choice. Maybe some monothalamid taxa happily live in the sand, but hard shelled foraminifers (Tubo- and Globothalamea) will avoid these environments and prefer to live in mud. Therefore your sampling site is hard to compare with mud environments of the other foram studies. You speak about that later, but may already discuss this here.
l. 506: we use instead of we uses
l. 520-523: and mudflats… where you will find most of intertidal forams. Could you add references for this environment?

Experimental design

no comment

Validity of the findings

no comment

Additional comments

no comment

Reviewer 3 ·

Basic reporting

I am satisfied by the modifications and improvements made by the authors on this new version of the manuscript. Therefore I accept the article for publication in PeerJ.

Experimental design

.

Validity of the findings

.

---

## Round 0.3 · accepted · Accept

I applaud the authors for following through with the comments and suggestions of the reviewers. The current version of the manuscript adds value to the molecular taxonomy of Foraminiferal communities and highlights the feasibility of COI as the metabarcoding sequences for this phylum.